# Partial heart transplantation for pediatric heart valve dysfunction: A clinical trial protocol

**Taufiek Konrad Rajab[1]***, **Brielle Ochoa**[1], **Kasparas Zilinskas**[2], **Jennie Kwon[1]**, **Carolyn L. Taylor[3]**, **Heather T. Henderson[3]**, **Andrew J. Savage[3]**, **Minoo Kavarana[1]**, **Joseph W. Turek[4]**, **John M. Costello[3]**

1 Division of Pediatric Cardiothoracic Surgery, Department of Surgery, Medical University of South Carolina, Charleston, South Carolina, United States of America, 2 College of Medicine, Medical University of South Carolina, Charleston, South Carolina, United States of America, 3 Division of Pediatric Cardiology, Department of Pediatrics, Medical University of South Carolina, Charleston, South Carolina, United States of America, 4 Department of Surgery, Duke University Hospitals, Durham, North Carolina, United States of America

* rajabt@musc.edu

**Data Availability Statement:** No datasets were generated or analysed during the current study. All relevant data from this study will be made available upon study completion.

## Abstract

Congenital heart defects are the most common type of birth defects in humans and frequently involve heart valve dysfunction. The current treatment for unrepairable heart valves involves valve replacement with an implant, Ross pulmonary autotransplantation, or conventional orthotopic heart transplantation. Although these treatments are appropriate for older children and adults, they do not result in the same efficacy and durability in infants and young children for several reasons. Heart valve implants do not grow with the. Ross pulmonary autotransplants have a high mortality rate in neonates and are not feasible if the pulmonary valve is dysfunctional or absent. Furthermore, orthotopic heart transplants invariably fail from ventricular dysfunction over time. Therefore, the treatment of irreparable heart valves in infants and young children remains an unsolved problem. The objective of this single-arm, prospective study is to offer an alternative solution based on a new type of transplant, which we call "partial heart transplantation." Partial heart transplantation differs from conventional orthotopic heart transplantation because only the part of the heart containing the heart valve is transplanted. Similar to Ross pulmonary autotransplants and conventional orthotopic heart transplants, partial heart transplants contain live cells that should allow it to grow with the recipient child. Therefore, partial heart transplants will require immunosuppression. The risks from immunosuppression can be managed, as seen in conventional orthotopic heart transplant recipients. Stopping immunosuppression will simply turn the growing partial heart transplant into a non-growing homovital homograft. Once this homograft deteriorates, it can be replaced with a durable adult-sized mechanical implant. The protocol for our single-arm trial is described. The ClinicalTrials.gov trial registration number is NCT05372757.

**Funding:** This research was supported by grants to TKR from the American Association of Thoracic Surgery, the Brett Boyer Foundation, the Children's Excellence Fund held by the Department of Pediatrics at the Medical University of South Carolina, the Saving tiny Hearts Society, the Emerson Rose Heart Foundation, the South Carolina Clinical & Translational Research Institute (NIH/NCATS Grant Number UL1 TR001450), and Philanthropy by Senator Paul Campbell. The funders did not and will not have a role in study design, data collection and analysis, decision to publish, or preparation of the manuscript.

**Competing interests:** The authors have declared that no competing interests exist.

## Introduction

Congenital heart defects are the most common type of birth defects in humans. In North America, 7 in 1000 live born children are affected [1]. Worldwide, this causes over 180,000 infant deaths per year [2]. Treatment of congenital heart defects frequently involves heart valve replacement. However, infants and young children who undergo valve replacements uniformly outgrow their valve implants, which always become dysfunctional over time due to the development of stenosis, regurgitation, or a combination of the two. As a result, serial cardiac reoperations to exchange the valve implants for successively larger versions are necessary as infants and children grow [3]. Additionally, current procedures in infants and neonates, such as the Ross procedure, have been known to cause substantial adverse effects. The Ross procedure has particularly high mortality in neonates [4, 5]. Children aged under one year who had undergone the Ross procedure also had a higher frequency of re-intervention postoperatively [4].

The current standard of care for replacing semilunar heart valves with an implant in neonates, infants, and very young children are homografts, which can be antibiotic-preserved or cryopreserved [6]. Homografts are sourced from human tissue donors and have been used as aortic valve substitutes for over half a century [7]. In the past, freshly procured valves from autopsy were sterilized in antibiotic solutions and used without further processing. These valves contain viable cells and are called homovital valves. Antibiotic-preserved valves that are removed from heart transplant recipients at the time of heart transplantation and stored in tissue culture medium at 4ºC contain 60% to 70% viable cells [8]. These valves can be used for up to six weeks. In contrast, cryopreserved and processed homografts, which do not contain viable cells, can be stored far longer and are commercially available. Therefore, the vast majority of homografts used today are cryopreserved. Prior data exists regarding the durability of homovital valves. A large series of 275 homovital aortic valves had actuarial rates for freedom from degenerative valve failure of 94% (+/- 2%) at 5 years and 89% (+/- 3%) at 10 years diagnosed at operation, by postmortem examination, or by routine echocardiography [9]. These valves were from donors aged 10 to 66 years and implanted in recipients aged 1.5 to 79 years. Unlike the proposed trial, these recipients were not immunosuppressed; therefore, no valve growth was expected. Recipients of these homovital aortic valve homografts are known to produce specific antibodies to human leukocyte antigens (HLA) determinants present in the homovital graft. However, analysis of HLA matching between homovital graft and recipients showed that the degree of HLA mismatch was not associated with homograft valve function [10]. Moreover, a case report from 1997 described a homovital graft harvested from a living donor and transplanted into a 54-year-old male within 20 minutes without any immunosuppressive treatment. Despite mismatch by ABO and tissue type, there was no clinical immune rejection and good echocardiographic function on follow-up nine months postoperatively [11]. Another trial examined the effect of the immunosuppressant mycophenolate mofetil (MMF) on HLA antibody response after homograft implantation in eight children aged 7 years or older. This study showed that MMF markedly decreases the HLA class I antibody response at one and three months post implantation [12]. However, the homografts were cryopreserved and thus did not contain viable cells and did not have the ability to grow.

Partial heart transplantation contributes a new approach to solve the problems with the current management for valve dysfunction in neonates, infants and young children [13]. Partial heart transplantation uses living homografts. Living homografts differ from conventional homografts in three important aspects: (1) The valves utilized in partial heart transplants will be procured to minimize the ischemic time just like conventional heart transplants, whereas homovital homografts are procured from donors at autopsy after a long period of ischemic time.

(2) Partial heart transplants will be implanted within six hours (similar to conventional orthotopic heart transplants), whereas the ischemia time for homografts is typically much longer.

(3) Recipients of a partial heart transplant will be immunosuppressed in the same way as conventional orthotopic heart transplant recipients, whereas recipients of a homograft are not typically immunosuppressed.

These differences should preserve viability of the partial heart transplant and allow it to grow, similar to the growth of a conventional orthotopic heart transplant.

Our central hypothesis for this trial is that transplantation of a living homograft ("partial heart transplantation") will be associated with superior outcomes compared to currently available implants (homografts, bioprostheses, mechanical valves, non-valved conduits) in infants and young children. Partial heart transplants may replace either a single semilunar valve or both semilunar valves depending upon the patient's needs. We will test our central hypothesis, and thereby achieve the objective of this trial by studying the following specific aims.

The primary aim is the feasibility and the safety of partial heart transplantation. Complication rates are expected to be similar or better to those of young children who have undergone orthotopic heart transplantation [14]. Secondary aims to be studied are valve annulus growth, stenosis, and regurgitation. We anticipate that stenosis and regurgitation will be minimal, and transplanted valve function will be similar to semilunar valve function observed in conventional heart transplants [15].

## Materials and methods

### Study design

A prospective nonrandomized, single-center, single arm pilot trial will be performed in infants and young children who require semilunar heart valve replacement. The study received approval from the Medical University of South Carolina (MUSC) Institutional Review Board Protocol Record 00114653 and the protocol was registered and approved on the ClinicalTrials.gov platform, trial number NCT05372757. The schedule of enrolment, interventions and assessments is shown in Fig 1.

### Study population

Five patients in need of a heart valve replacement will be included in the study. A study investigator who normally participates in our biweekly pediatric heart center case conference meeting and has access to patient records and surgical referrals for valve replacement will identify potential candidates for the trial. Study investigators will obtain permission from the primary cardiologist or cardiac surgeon prior to approaching parents or legal guardians to discuss participation in the trial. Telemedicine may be utilized to initially discuss the risks and benefits associated with the study with parents or legal guardians of potential subjects who are outpatients, but an in-person meeting will be required for a formal consent discussion. Expectant mothers who are carrying fetuses that are expected to require heart surgery requiring a valve replacement within the first two years of life may be approached for enrollment in the study as well.

The interventional portion of the study will take place in the cardiac operating rooms and in the Pediatric Cardiac Intensive Care Unit at Shawn Jenkins Children's Hospital at MUSC. Additional observational data will be collected pre- and intraoperatively, and postoperatively on the Cardiac Stepdown Unit and in the outpatient cardiology clinics at MUSC's Summey Medical Pavilion.

| TIMEPOINT** | Enrolment | Allocation | Post-allocation | | | | | Close-out |
|---|---|---|---|---|---|---|---|---|
| | $-t_1$ | 0 | $t_1$ | $t_2$ | $t_3$ | $t_4$ | etc. | $t_x$ |
| **ENROLMENT:** | | | | | | | | |
| **Eligibility screen** | X | | | | | | | |
| **Informed consent** | X | | | | | | | |
| **Allocation** | | X | | | | | | |
| **INTERVENTIONS:** | | | | | | | | |
| *Partial heart transplant* | | | X | | | | | |
| **ASSESSMENTS:** | | | | | | | | |
| *Medical history, physical examination, investigations (including echocardiogram)* | X | X | | | | | | |
| *Feasibility of partial heart transplantation* | | | X | | | | | |
| *Safety of partial heart transplantation* | | | | X | X | X | etc. | X |
| *Valve growth, stenosis and regurgitation* | | | | X | X | X | etc. | X |

**Fig 1. Schedule of enrolment, interventions, and assessments.** The figure shows Standard Protocol Items: Recommendations for Interventional Trials (SPIRIT) schedule of enrolment, interventions and assessments information. **Post-allocation time points occur at 6 months intervals, starting when the first subject undergoes partial heart transplantation and continuing until one year after all subjects have been enrolled.

To be eligible to participate in this study, the patients can be either male or female and must meet the following inclusion and exclusion criteria:

Inclusion Criteria

- Children less than 2 years of age who are referred for a cardiac operation that involves a primary semilunar valve replacement, or children less than 2 years of age who are referred for a cardiac operation involves an initial replacement of a previously placed prior homograft, bioprosthetic, or mechanical valve in the aortic or pulmonary position

- Deemed acceptable for partial heart transplantation based on the standard evaluation process used for conventional orthotopic heart transplantation at our center (see S1 Appendix)

- Have insurance approval

- Provide written informed consent of both parents or legal guardians; if there is only one parent or legal guardian, consent from that individual will be adequate.

  Exclusion Criteria

- Absolute contraindications for conventional orthotopic heart transplantation:

  - Severe bilateral long segment pulmonary arterial hypoplasia

  - Bilateral pulmonary vein stenosis

  - <34 weeks corrected gestational age

  - Persistent acidosis with a pH <7.1

- Diagnosis of immune deficiency

- Infectious disease

  - Active sepsis

  - Hepatitis B surface antigenemia

  - HIV positivity

- Inability for the parent(s) or legal guardian(s) to understand English or Spanish

- Failure to pass the following psychosocial evaluation:

  - The candidate should reside within four hours ground traveling time from Medical University of South Carolina for a minimum of four to six months post-transplantation to facilitate close follow-up.

  - The candidate's family should be capable of long-term supportive care of the child and be able to support the medical needs of the candidate in follow-up.

  - Parental (custodial) alcohol and/or substance abuse.

  - Documented parental (custodial) child abuse or neglect.

  - Parent (custodian) with cognitive/psychiatric impairment severe enough to limit comprehension of medical regimen.

## Treatment

The evaluations, tests, and procedures below are typically performed as standard clinical care for recipients of a conventional heart transplant or heart valve replacement. No additional study-specific tests or procedures are planned as part of this trial. We will obtain all the information necessary for the research from the electronic medical record as would be consistent with standard practice for a conventional heart transplant or heart valve replacement.

**Pre-transplantation evaluation.** An independent determination of whether a patient is a candidate for partial heart transplantation will be made in a standing multi-disciplinary conference that includes the pediatric heart transplant team, pediatric cardiac surgery team, and pediatric cardiology team, as is the clinical standard for orthotopic heart transplants. Required information for each potential participant should include but is not limited to the following:

- History

- Demographics

- Past medical history including cardiac diagnoses and interventions

- Pre-sensitization events such as transfusions

- Dates and types of immunizations

- Past surgical history, including cardiac surgery or transplant

- Medications

- Allergies

- Social history

- Physical examination

- Laboratory tests, including but not limited to:

  - ABO type

  - Panel Reactive Antibody (PRA)

  - Pretransplant infectious diseases screening

  - Complete blood count

  - Cytomegalovirus (CMV) titers (IgG, IgM)

  - Epstein-Barr virus (EBV) titers (EBV-IgG, EBV-IgM)

  - Human Immunodeficiency Virus testing

  - Rapid Plasma Reagin or equivalent

  - Hepatitis B Surface antigen

  - Hepatitis C antibody

- Echocardiogram

- Chest radiograph

Research data will be collected from the medical record, history, and preoperative echocardiogram.

**Preoperative procedures.** Written informed consent for partial heart transplantation will be obtained from the parents or legal guardians of each subject. Potential subjects and their parents or legal guardians will undergo a full evaluation for candidacy for partial heart transplant using well-established criteria and workflows for infants and young children who are being assessed for conventional heart transplantation. These patients will be listed for partial heart transplant using pre-established mechanisms similar to conventional heart transplantation via the regional South Carolina organ procurement network, We Are Sharing Hope.

For a subject to be enrolled in the trial, a time window will be established by an attending pediatric cardiologist as to how long the subject can be listed for partial heart transplantation before a conventional valve replacement option is scheduled. This time window will be determined based on the subject's clinical status and severity of valve dysfunction and may be revised over time based on evolving clinical status.

As is the case for patients undergoing conventional heart transplantation, donor hearts will be selected based on the recipient's ABO blood group, PRA, weight, CMV status, and EBV

status. Parents or legal guardians of enrolled subjects will be required to carry a transplant pager, consistent with established practice for patients who are listed for conventional heart transplantation.

When a donor heart becomes available, a focused echocardiogram will be performed on the donor heart to specifically assess the semilunar valve annulus size and function before the heart is accepted. The donor heart will be procured and transported to MUSC by our cardiac surgeons using practices consistent with those utilized for conventional heart transplantation. The donor semilunar valve will be excised from the donor heart by a pediatric cardiac surgeon. The heart valve will be accepted after it is visually inspected by the procurement team at the donor hospital.

**Operative procedures.** Subjects will be scheduled for an emergent operation using established mechanisms. The recipient operation will not start until the donor valve is visually inspected and accepted by the MUSC procurement team. When the donor heart valve arrives in the MUSC operating room, the usual procedure for organ check-in will be followed (S1 Appendix).

The donor valve will be implanted into the recipient using techniques identical to those used for homograft semilunar valve replacement [16]. If for any reason the donor heart valve is unsuitable, a cryopreserved homograft valve will be implanted. Consistent with standard practice, a transesophageal or epicardial echocardiogram will be obtained in the operating room to assess valve and myocardial function before the sternotomy incision is closed.

Immunosuppression will be administered intraoperatively consistent with practices utilized for conventional heart transplantation, which typically includes administration of a calcineurin inhibitor such as tacrolimus, mycophenolate mofetil (MMF), and steroids [17].

Research data will be collected from the medical record, including the operative note (surgeon, surgical procedure, cardiopulmonary bypass time, cross-clamp time, ischemic time), and intraoperative echocardiogram (size and function of both the excised and newly implanted donor valve).

**Early postoperative procedures.** The patient will be admitted to the pediatric cardiac intensive care unit (PCICU) following the operation. Routine postoperative care, including but not limited to monitoring, the use of mechanical ventilation, inotropic infusions, narcotics, and sedative agents, will be provided consistent with established institutional practices. Antibiotic prophylaxis will be administered consistent with institutional practices used for patients recovering from conventional heart transplantation. Immunosuppression will again be administered and monitored post operatively consistent with practices utilized for orthotopic heart transplantation, as previously described.

A pediatric cardiologist will round on the patient at regular intervals and be available to guide postoperative management. Transthoracic echocardiograms will be obtained postoperatively at intervals consistent with those used for patients recovering from conventional heart transplantation. The patient will be discharged from the hospital once all standard criteria have been met and when deemed safe by the attending pediatric cardiac surgeon, attending pediatric cardiologist, and attending heart transplant physician.

Data recorded in the MUSC medical record will be collected for research, including inotrope requirement, time of extubation, arrhythmias and treatments, PCICU discharge, PCICU readmissions, interventions during the same hospitalization, infections, incidence and severity for acute kidney injury, adverse events, and date of discharge. For the patient who lives out of the greater Charleston area and do not follow-up at MUSC, the research team will contact the primary clinical cardiologist to obtain medical record information.

**Late postoperative procedures.** Patients who reside outside of the greater Charleston area will be required to stay locally for six weeks at the Ronald McDonald House or elsewhere

following hospital discharge, consistent with practice for patients recovering from conventional heart transplantation. Patients will receive follow-up care in our outpatient pediatric heart transplant clinic at intervals consistent with those used for infants and young children recovering from conventional heart transplantation.

Once deemed appropriate by the pediatric heart transplant attending, patients who reside outside of the greater Charleston area may have some of their outpatient cardiology clinic visits conducted by a local cardiologist. In such cases, communication between the pediatric heart transplant attending of the local cardiologist will occur consistent with established practices for children who have undergone conventional heart transplantation. Immunosuppression will be maintained at standard levels given in conventional heart transplantation for at least the first year following valve transplant.

Once patients have grown and further valve growth is no longer required, consideration may be given to weaning or discontinuing immunosuppression. Once immunosuppression is stopped the heart valve transplant will turn into a homograft and stop growing. We anticipate that once the homograft deteriorates, it may be replaced with a prosthetic valve. Follow-up by the research team will occur biannually in conjunction with clinically indicated visits until one year after the transplanted valve has been replaced.

If a patient dies, parents or legal guardians will be offered an autopsy. The aim of the autopsy should be not only to establish the cause of death as accurately as possible, but also to evaluate the effects of partial heart transplantation and to detect and delineate any intercurrent processes, both known and unsuspected.

## Data management methods

Data will be collected at intervals as described above. Descriptions of confidentiality provisions and data safety monitoring boards (DSMB) are provided in a later section. In order to preserve the confidentiality of the participants, the following interventions will be undertaken:

A. Case report forms (CRFs) will be developed prior to the onset of the trial and completed by a study investigator for each patient.

 a. An enrollment form will be completed once written informed consent is provided by the parents or legal guardians.

 b. Adverse event forms will be completed as needed throughout the operative hospitalization.

B. A dedicated Excel database will be developed for the trial.

 a. All data from the CRFs will be entered into a dedicated Excel database by a study team member.

C. In the case of a subject withdrawing from the trial, the data will be potentially included in the final analysis depending on the time enrolled in the trial.

Missing data will be addressed by omission or analysis as appropriate. Sensitivity analysis will be planned to avoid misleading inference when complete-case analysis is performed.

## Quality control methods

The principal investigator or study coordinator will review all CRFs and data within the study database for completeness and accuracy on an annual basis. Recruitment will also be monitored on an annual basis.

## Assessment of results

The primary outcomes will be the feasibility and safety of partial heart transplantation. The feasibility will be determined by the ability to perform the new operation in at least one patient per year for three years after enrollment opens. The safety of the procedure will be indicated by survival and re-interventions that are the same or are better than those reported in the literature for patients of similar age undergoing conventional valve replacement. Safety will primarily be defined as patient survival or re-operation or transcatheter reintervention on the transplanted valve at six-month intervals, starting six months after the first subject undergoes a valve transplant, and continuing until one year after all five subjects have been enrolled. Safety will also be assessed by tracking rates of immunosuppression side effects, including development of insulin-dependent diabetes, chronic renal insufficiency, and significant infections.

The secondary outcomes will be valve annulus growth over time, stenosis and regurgitation. The valve annulus diameter will be measured using echocardiography in two planes at baseline and every six months following the operation. The valve annulus Z-score should remain relatively unchanged over time, indicating an increase in valve annulus growth corresponding with an increase in body surface area. This change should correspond to the growth of conventional heart transplants and Ross pulmonary autografts [18–20]. Valve stenosis and regurgitation will be assessed via transthoracic echocardiogram with pulse Doppler and with color Doppler, respectively, every six months following the operation.

## Possible risks to participants

This study involves greater than minimal risk with the potential for direct benefit. In conventional heart transplantation, the most serious risks are graft failure, cardiac allograft vasculopathy, infection and acute rejection, which altogether account for the majority of deaths after conventional heart transplantation. The relative importance of these causes of death vary over time post-transplant. Risks of partial heart transplantation are expected to be similar to the risks of standard valve replacement but less than the risks of conventional heart transplant. A complete list of risks is provided in S2 Appendix.

1. The most serious risks are scheduling issues related to donor heart availability and rejection of the transplanted valve. The risk of rejection is theoretically lower compared to a full heart transplant. The risk of death during and after the operation is similar to the risks for standard valve replacement with a homograft. There is no risk of disease developing in the blood vessels that supply the heart (i.e., coronary arteries) and no risk of heart muscle dysfunction given that these tissues are not being transplanted.

2. The surgery will result in a permanent scar in the middle of the chest. This is the same scar that would result from a standard valve replacement or full heart transplant. This scar may require medical follow-up after surgery and may temporarily restrict the subject from exercise and/or sports participation in the future.

3. Open heart surgery involves the risk of death, brain damage, or other organ damage such as kidney injury.

4. The need for blood transfusion during surgery poses a risk of infection from the blood or a reaction to the blood. The risk of infection from transfused blood is extremely low given current screening processes.

5. Patients receiving a partial heart transplant may also become sensitized to the transplanted valve tissue. This will decrease the donor pool if the patient ever requires a heart transplant.

6. There is a risk of loss of confidentiality.

7. There may also be additional risks that are currently unknown.

## Possible benefits to participants

The possible benefits to participants may include the need for fewer valve replacement operations and their substantial morbidity and mortality during childhood.

1. The most important advantage of partial heart transplantation is growth of the transplanted valve. This minimizes the morbidity and mortality from successive implant exchanges.

2. Avoiding reoperations will improve neurodevelopmental outcomes as repeat cardiac operations are associated with worse neurodevelopmental outcomes [21].

3. Avoiding reoperations will decrease psychological and financial stress for subjects' caregivers and families [22].

4. The transplanted valve would be durable without the need for anticoagulation.

5. The transplanted valve may have better valve function with more favorable loading conditions on the ventricular myocardium.

6. Because of fewer surgeries and their risks, the subjects may potentially have longer lifespans.

## Data analysis

Data will be exported from the study database for statistical analysis. Given the goal sample size of five participants and lack of control group, simple descriptive statistics will be used.

## Confidentiality provisions

All completed CRFs will remain confidential and will be stored in locked cabinets within the locked offices of the study investigators at MUSC, accessible only to study team members. Study data will reside in password-protected computers within locked offices of study investigators at MUSC, accessible only to study team members. All data stored on MUSC servers are password-protected and backed up nightly. The network is protected on multiple levels that include firewalls, intrusion detection software, and continuous monitoring by the Information Services Security Department.

The study investigators will be primarily responsible for monitoring all study patients for the occurrence of adverse events and unanticipated problems as outlined above.

A DSMB will be chaired by a physician and will be used to monitor the progress of this clinical trial and review safety and effectiveness data while the trial is ongoing. DSMB members will be comprised of senior faculty from other institutions, and will include a pediatric transplant cardiologist, a pediatric cardiac intensivist, and a pediatric cardiac surgeon. None of these individuals will be directly involved with the undertaking of this study or affiliated with MUSC.

The DSMB will be responsible for reviewing the research protocol (and any amendments thereof) and the informed consent documents (S3 Appendix) prior to the onset of patient recruitment. The DSMB will meet virtually one month after each of the first three patients are enrolled and annually thereafter to review study data. The study investigators will prepare interim reports, including a summary of outcomes and adverse events, for DSMB review.

The DSMB will consider and review the following: major protocol modifications proposed by the study team; participant recruitment, accrual and retention; confidentiality issues; factors external to the study when relevant, such as scientific or therapeutic developments that may have an impact on the safety of the subjects or the ethics of the trial; and investigator-assigned relationships between the intervention and adverse events. Following each meeting, the DSMB will prepare a written report on the safety and scientific progress of the trial and make recommendations to the study investigators concerning continuation, termination or modification of the trial based on the observed beneficial or adverse effects of the research. The DSMB will also meet at the conclusion of the study, and as needed to review serious adverse events.

A subject will be removed from the study if a parent or legal guardian withdraws consent, or if one of the patient's primary attending physicians elects to remove the patient from the study. Additionally, all clinicians who are uninvolved with the trial but participate in the care of the subjects will be encouraged to report adverse events to the study investigators. The study investigators will meet every six months to review adverse events and to assess the risk/benefit ratio for continuing the trial.

Any action that results in a temporary or permanent suspension of the trial will be reported as soon as possible by the investigators to the MUSC IRB and DSMB. If the DSMB or IRB requests a temporary or permanent suspension of the trial, they will notify the principal investigator who will promptly notify all co-investigators and the study team. Annually, the principal investigator and study coordinator will meet for data verification and protocol compliance checks. Specifically, these individuals will monitor whether all enrolled subjects met entry criteria, whether any deviations from the study protocol occurred, and whether any patients were prematurely removed from the protocol by managing physicians or withdrawal of parental consent. The principal investigator or study coordinator will review all CRFs and data within the study database for completeness and accuracy on an annual basis.

## Discussion

Partial heart transplantation for valve disorders is innovative because it is new and substantially different from current approaches and opens new horizons for the management of infants and young children who require heart valve replacements. This contribution is significant because infants and young children with irreparable valves will be spared morbidity from successive implant exchanges and, in the case of mechanical valve replacement, from therapeutic anticoagulation. Partial heart transplantation can be performed using donor hearts with poor ventricular function and slow progression to donation after cardiac death. This should ameliorate donor heart utilization and avoid both primary orthotopic heart transplantation in children with unrepairable heart valve dysfunction and progression of these children to end-stage heart failure [23].

Partial heart transplantation is not without risk. Valve replacement operations in infants and young children entail inherent risk for morbidity and mortality regardless of the type of implant. Patients undergoing a partial heart transplantation will receive immunosuppression until the transplanted valve can be replaced with a conventional adult-sized implant. Importantly, modern immunosuppression is well tolerated and pediatric heart transplants have some of the best outcomes of any solid organ transplant groups. Lower levels of immunosuppression are hypothetically feasible because clinical rejection of conventional heart transplants does not typically affect semilunar heart valve function [24]. Stopping immunosuppression would transform the partial heart transplant into a homograft, which is the current standard of care for heart valve replacement with an implant in neonates, infants and young children [11].

In summary, this evaluation of partial heart transplantation in this single-arm clinical trial has the potential to create a paradigm shift in the management of pediatric valve disorders. In the future, extensive statistical analysis based on a larger sample will be necessary to further evaluate partial heart transplantation.

## Supporting information

**S1 Appendix. Orthotopic heart transplant evaluation.**
(DOCX)

**S2 Appendix. Complete list of risks.**
(DOCX)

**S3 Appendix.**
(DOCX)

**S1 Checklist.**
(DOCX)

**S1 File.**
(PDF)

## Author Contributions

**Conceptualization:** Taufiek Konrad Rajab.

**Funding acquisition:** Taufiek Konrad Rajab.

**Methodology:** Taufiek Konrad Rajab, Carolyn L. Taylor, Heather T. Henderson, Andrew J. Savage, Minoo Kavarana, Joseph W. Turek, John M. Costello.

**Project administration:** Taufiek Konrad Rajab, John M. Costello.

**Supervision:** Taufiek Konrad Rajab.

**Validation:** Taufiek Konrad Rajab.

**Writing – original draft:** Taufiek Konrad Rajab, John M. Costello.

**Writing – review & editing:** Taufiek Konrad Rajab, Brielle Ochoa, Kasparas Zilinskas, Jennie Kwon, Joseph W. Turek, John M. Costello.

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
