## [Decision Letter · Decision Letter 0]

7 Oct 2022

PONE-D-22-23111Partial Heart Transplantation for Severe Pediatric Semilunar Valve Dysfunction: A Clinical Trial ProtocolPLOS ONE

Dear Dr. Zilinskas,

Thank you for submitting your manuscript to PLOS ONE. After careful consideration, we feel that it does not fully meet PLOS ONE’s publication criteria as it currently stands. Therefore, we invite you to submit a revised version of the manuscript that addresses the points raised during the review process.

We look forward to receiving your revised manuscript.

Kind regards,

Jaimin R. Trivedi, MBBS, MPH

Academic Editor

PLOS ONE

Journal Requirements:

"The funders had and will not have a role in study design, data collection and analysis, decision to publish, or preparation of the manuscript."

"BG and JK are funded by NIH-NHLBI Institutional Postdoctoral Training Grants (T32 HL-007260). TKR is funded by the American Association for Thoracic Surgery, the Children's Excellence Fund held by the Department of Pediatrics at the Medical University of South Carolina, the Brett Boyer Foundation, and the Emerson Heart Foundation."

We note that you have provided funding information that is not currently declared in your Funding Statement. However, funding information should not appear in the Funding section or other areas of your manuscript. We will only publish funding information present in the Funding Statement section of the online submission form. 

"The funders had and will not have a role in study design, data collection and analysis, decision to publish, or preparation of the manuscript."

4. We note that the original protocol that you have uploaded as a Supporting Information file contains an institutional logo. As this logo is likely copyrighted, we ask that you please remove it from this file and upload an updated version upon resubmission.

Additional Editor Comments (if provided):

The manuscript has been reviewed by independent reviewers including a statistical reviewer. They have raised some important points including certain ethical considerations and statistical/study design suggestions. Please carefully consider their comments and suggestions.

Reviewers' comments:

Reviewer's Responses to Questions

**Comments to the Author**

1. Does the manuscript provide a valid rationale for the proposed study, with clearly identified and justified research questions?

Reviewer #1: Yes

Reviewer #2: Yes

2. Is the protocol technically sound and planned in a manner that will lead to a meaningful outcome and allow testing the stated hypotheses?

Reviewer #1: Yes

Reviewer #2: Partly

3. Is the methodology feasible and described in sufficient detail to allow the work to be replicable?

Reviewer #1: Yes

Reviewer #2: Yes

4. Have the authors described where all data underlying the findings will be made available when the study is complete?

Reviewer #1: Yes

Reviewer #2: No

5. Is the manuscript presented in an intelligible fashion and written in standard English?

Reviewer #1: Yes

Reviewer #2: Yes

6. Review Comments to the Author

You may also provide optional suggestions and comments to authors that they might find helpful in planning their study.

Reviewer #1: Ochoa and colleagues describe a protocol for a prospective nonrandomized, single-center, single arm pilot trial where a “partial heart transplant” will be performed in infants and young children with semilunar valve disease. The hypothesis is that the transplant semilunar valves will grow with the patient and avoid known complications associated with current valve options for these children. This a reasonable protocol, however, I have following specific comments for the authors

“Inability for the parent(s) or legal guardian(s) to understand English or Spanish”, this seems like an inappropriate exclusion criterion, especially when all study procedures are standard clinical care, and one would expect that inability to understand these languages does not preclude clinical care.

There is another ethical issue/potential risk with this procedure that is not discussed. A potential organ for life-saving heart transplant is diverted for a treatment that has other surgical options. This should be discussed and potentially be part of the informed consent. Furthermore, donor selection is not discussed in detail, could these donors have suboptimal cardiac function and not suitable for conventional heart transplant?

It may be after this submission, but there has been a clinical case described as “first partial heart transplant” from Duke University in the news lately. Citation of that would be important for complete reference of prior work.

Lower levels of immunosuppression are proposed as a possibility after first year, a description of such protocol is important for completeness.

Reviewer #2: This study addresses an interesting topic. It is well detailed and may lead to interesting results. Some comments follow.

1. The sample size is rather small. It is necessary to specify the power of the study. Similarly, the fact that it is a nonrandomized single-center study may strongly limit its practical usefulness. Indeed, my main concern is about how general the obtained results would be. It is difficult to understand why such a small sample size is aimed to be collected.

2. Missing data management needs a better discussion. Missing data could be completely-at-random, at-random, not-at-random. Sensitivity analysis should be planned to avoid misleading inference whenever complete-case analysis is performed.

3. The data analysis is based on descriptive statistics only. This strongly limits the project. If research questions cannot be tested, if inference cannot be performed, the study simply describes a specific sample, that may not be representative of a larger population. Which general info do you expect to get from a simple descriptive analysis? Cause-effect relationships cannot be investigated, nor any characterization of the outcomes could be discussed. The data analysis is not sufficient to provide informative results. I strongly suggest to plan a more detailed and extensive statistical analysis, based on a larger sample. Statistical inference must be guranteed.

7. PLOS authors have the option to publish the peer review history of their article (what does this mean?). If published, this will include your full peer review and any attached files.

Reviewer #1: No

Reviewer #2: No

---

## [Author Response · Author response to Decision Letter 0]

22 Nov 2022

Reviewer #1: Ochoa and colleagues describe a protocol for a prospective nonrandomized, single-center, single arm pilot trial where a “partial heart transplant” will be performed in infants and young children with semilunar valve disease. The hypothesis is that the transplant semilunar valves will grow with the patient and avoid known complications associated with current valve options for these children. This a reasonable protocol, however, I have following specific comments for the authors

“Inability for the parent(s) or legal guardian(s) to understand English or Spanish”, this seems like an inappropriate exclusion criterion, especially when all study procedures are standard clinical care, and one would expect that inability to understand these languages does not preclude clinical care.

RESPONSE: This exclusion criterion was added because we do not have the ability to translate the clinical trial consent form accurately into other languages.

There is another ethical issue/potential risk with this procedure that is not discussed. A potential organ for life-saving heart transplant is diverted for a treatment that has other surgical options. This should be discussed and potentially be part of the informed consent. Furthermore, donor selection is not discussed in detail, could these donors have suboptimal cardiac function and not suitable for conventional heart transplant?

RESPONSE: We agree that donor selection is important and needs to be discussed. We also agree that donors with suboptimal cardiac function and not suitable for conventional transplant would be ideal. We added the following information to the discussion: “Partial heart transplantation can be performed using donor hearts with poor ventricular function and slow progression to donation after cardiac death. This should ameliorate donor heart utilization and avoid both primary orthotopic heart transplantation in children with unrepairable heart valve dysfunction and progression of these children to end-stage heart failure.” We also cited a recent review that we wrote about this very important topic (Sherard et al. Partial heart transplantation can ameliorate donor organ utilization. J Card Surg . 2022 Oct 19. doi: 10.1111/jocs.17050. Online ahead of print.). 

It may be after this submission, but there has been a clinical case described as “first partial heart transplant” from Duke University in the news lately. Citation of that would be important for complete reference of prior work.

RESPONSE: this case is not yet published in the scientific literature.

Lower levels of immunosuppression are proposed as a possibility after first year, a description of such protocol is important for completeness.

RESPONSE: This is beyond the scope of the current clinical trial. Therefore, the reference to lower levels of immunosuppression was deleted.

Reviewer #2: This study addresses an interesting topic. It is well detailed and may lead to interesting results. Some comments follow.

1. The sample size is rather small. It is necessary to specify the power of the study. Similarly, the fact that it is a nonrandomized single-center study may strongly limit its practical usefulness. Indeed, my main concern is about how general the obtained results would be. It is difficult to understand why such a small sample size is aimed to be collected.

RESPONSE: The reason for the small sample size is the cost of the operation. Partial heart transplantation involves procurement of the donor heart using jet airplane transport in order to minimize the ischemia time. Therefore a larger number of patients for this pilot trial would be prohibitive.

2. Missing data management needs a better discussion. Missing data could be completely-at-random, at-random, not-at-random. Sensitivity analysis should be planned to avoid misleading inference whenever complete-case analysis is performed.

RESPONSE: We agree. The following paragraph was inserted in the data management methods section: “Missing data will be addressed by omission or analysis as appropriate. Sensitivity analysis will be planned to avoid misleading inference when complete-case analysis is performed.”

3. The data analysis is based on descriptive statistics only. This strongly limits the project. If research questions cannot be tested, if inference cannot be performed, the study simply describes a specific sample, that may not be representative of a larger population. Which general info do you expect to get from a simple descriptive analysis? Cause-effect relationships cannot be investigated, nor any characterization of the outcomes could be discussed. The data analysis is not sufficient to provide informative results. I strongly suggest to plan a more detailed and extensive statistical analysis, based on a larger sample. Statistical inference must be guranteed.

RESPONSE: We agree that a more detailed extensive statistical analysis based on a larger sample would be helpful. We are planning this as the next step once funding for can be secured, either for insurance companies that already fund all expenses for conventional heart transplants or via a grant. The following sentence was added to the conclusion: “In the future, extensive statistical analysis based on a larger sample will be necessary to further evaluate partial heart transplantation.”

---

## [Decision Letter · Decision Letter 1]

21 Dec 2022

Partial Heart Transplantation for Severe Pediatric Semilunar Valve Dysfunction: A Clinical Trial Protocol

PONE-D-22-23111R1

Dear Dr. Rajab,

We’re pleased to inform you that your manuscript has been judged scientifically suitable for publication and will be formally accepted for publication once it meets all outstanding technical requirements.

Kind regards,

Jaimin R. Trivedi, MBBS, MPH

Academic Editor

PLOS ONE

Additional Editor Comments (optional):

Reviewers' comments:

Reviewer's Responses to Questions

**Comments to the Author**

1. Does the manuscript provide a valid rationale for the proposed study, with clearly identified and justified research questions?

Reviewer #1: Yes

Reviewer #2: Yes

2. Is the protocol technically sound and planned in a manner that will lead to a meaningful outcome and allow testing the stated hypotheses?

Reviewer #1: Yes

Reviewer #2: Yes

3. Is the methodology feasible and described in sufficient detail to allow the work to be replicable?

Reviewer #1: Yes

Reviewer #2: Yes

4. Have the authors described where all data underlying the findings will be made available when the study is complete?

Reviewer #1: Yes

Reviewer #2: Yes

5. Is the manuscript presented in an intelligible fashion and written in standard English?

Reviewer #1: Yes

Reviewer #2: Yes

6. Review Comments to the Author

You may also provide optional suggestions and comments to authors that they might find helpful in planning their study.

Reviewer #1: Authors have responded to the comments adequately and made appropriate changes. This manuscript is well written.

Reviewer #2: All my comments are properly addressed. I believe this work is ready to be published. It will be a starting point for further analyses on the topic.

7. PLOS authors have the option to publish the peer review history of their article (what does this mean?). If published, this will include your full peer review and any attached files.

Reviewer #1: No

Reviewer #2: No

---

## [Editor Report · Acceptance letter]

30 Jan 2023

PONE-D-22-23111R1 

Partial Heart Transplantation for Pediatric Heart Valve Dysfunction: A Clinical Trial Protocol 

Dear Dr. Rajab:

I'm pleased to inform you that your manuscript has been deemed suitable for publication in PLOS ONE. Congratulations! Your manuscript is now with our production department. 

Kind regards, 

on behalf of

Dr. Jaimin R. Trivedi 

Academic Editor

PLOS ONE